# A practical concept for catalytic carbonylations using carbon dioxide

Rui Sang [1,3], Yuya Hu[1,3], Rauf Razzaq[1,3], Guillaume Mollaert [2], Hanan Atia[1], Ursula Bentrup[1], Muhammad Sharif[1], Helfried Neumann [1], Henrik Junge [1] ✉, Ralf Jackstell [1] ✉, Bert U. W. Maes [2] ✉ & Matthias Beller [1] ✉

The rise of $CO_2$ in atmosphere is considered as the major reason for global warming. Therefore, $CO_2$ utilization has attracted more and more attention. Among those, using $CO_2$ as C1-feedstock for the chemical industry provides a solution. Here we show a two-step cascade process to perform catalytic carbonylations of olefins, alkynes, and aryl halides utilizing $CO_2$ and $H_2$. For the first step, a novel heterogeneous copper $10Cu@SiO_2$-PHM catalyst exhibits high selectivity (≥98%) and decent conversion (27%) in generating CO from reducing $CO_2$ with $H_2$. The generated CO is directly utilized without further purification in industrially important carbonylation reactions: hydroformylation, alkoxycarbonylation, and aminocarbonylation. Notably, various aldehydes, (unsaturated) esters and amides are obtained in high yields and chemo-/regio-selectivities at low temperature under ambient pressure. Our approach is of interest for continuous syntheses in drug discovery and organic synthesis to produce building blocks on reasonable scale utilizing $CO_2$.

The rise of anthropogenic carbon dioxide caused by fossil-based energy production is considered as the major reasons for global warming. Thus, finding solutions to reduce global $CO_2$ emissions into the atmosphere is of crucial importance for coming generations[1,2]. On the other hand, $CO_2$ provides the basis for the photosynthesis and represents an increasingly interesting C1-feedstock for the chemical industry. In this context, carbon dioxide, as the "waste carbon" from burning of fossil fuels, could be converted into "value carbon" in advanced chemicals[3–8]. However, the efficient chemical conversion of $CO_2$ is still challenging due to its thermodynamic stability[9]. By using strong reductants such as silanes or metals this problem can be overcome; however, significant amounts of waste are formed[10–12]. In contrast, the reduction of $CO_2$ with $H_2$ in the presence of specific catalysts allows to synthesize formic acid, methanol and even methane with water as the only by-product[13–15].

For industrial applications, the reduction of $CO_2$ with $H_2$ to selectively generate carbon monoxide via the reverse water-gas shift (RWGS) reaction is interesting as CO is an important building block in organic chemistry and the chemical industry[16,17]. Representative examples are hydroformylations to aldehydes[18], Monsanto as well as Cativa acetic acid syntheses[19], Eastman-Kodak acetic anhydride process[20], the synthesis of methyl methacrylate by Lucite alpha process[21], and others[22]. In addition, more special carbonylation reactions (Pauson-Khand, carbonylative Heck, Suzuki, Negishi, Tsuji reactions, etc.)[23–25] were invented in the past decades, which provide a toolbox for synthetic chemists to prepare structurally complex organic compounds including valuable agrochemicals, pharmaceuticals, and specialties. However, many synthetic organic chemists continue to be reluctant to perform carbonylations due to the toxicity and handling of CO. In fact, the use, storage, and transport of CO limits its utilization. For example, past efforts of Bayer AG in Germany to build a CO transport pipeline was not possible due to public protest and safety concerns (more information in CO transport pipeline problem of Bayer AG, see https://www.dw.com/en/bayer-subsidiary-holds-on-to-controversial-co-pipeline/a-38382768). Consequently, the development of alternative production of CO from less toxic compounds via in- or ex-situ continues to attract significant attention[26,27]. In this respect, the development of heterogeneous catalytic systems to

[1]Leibniz-Institut für Katalyse e.V., Albert-Einstein-Str. 29a, 18059 Rostock, Germany. [2]Organic Synthesis Division, Department of Chemistry, University of Antwerp, Groenenborgerlaan 171, 2020 Antwerp, Belgium. [3]These authors contributed equally: Rui Sang, Yuya Hu, Rauf Razzaq. ✉e-mail: Henrik.junge@catalysis.de; ralf.jackstell@catalysis.de; bert.maes@uantwerpen.be; matthias.beller@catalysis.de

promote RWGS reaction is interesting, too[28]. Apart from precious metals Rh, Pt, Pd, Au, also 3d-metals Ni, Cu, and Fe supported on oxides were reported in the $CO_2$ hydrogenation to CO[17,29–31]. However, the high conversion of $CO_2$ combined with selective formation of CO remains a major challenge. For instance, catalysts based on $Pd/TiO_2$, Pd/ZnO, $Pt/SiO_2$, or bimetallic catalysts such as $PtCo/TiO_2$ displayed high CO selectivity over methane and methanol, but lower $CO_2$ conversion was attained. Notably, Cu-based catalysts have been extensively studied, but most of them produced CO only in low to moderate selectivity and significant amounts of MeOH and/or methane are accompanied[29].

To reach high selectivity towards CO, multicomponent catalysts are essential[32]. Apparently, the dispersion, surface morphology, and particle size of such materials have to be carefully controlled to improve both activity and CO selectivity[17]. The RWGS reaction is an endothermic process; therefore, high reaction temperature should facilitate the formation of CO. However, currently known Cu-based catalysts are not suitable for operating at high temperature because they are easily deactivated by sintering[33].

In the past decade, the use of CO surrogates such as formates, aldehydes, metal carbonyls, COgen, and SilaCOgen has become increasingly popular, and many interesting applications have been achieved in organic synthesis[26,34–38]. Compared to all these works, CO generation from $CO_2$ and its direct follow-up use in carbonylation reactions has been largely neglected. Indeed, only a few chemical and electrochemical $CO_2$-to-CO conversion coupled with CO utilization for carbonylations were reported, until now. Chemically, Tominaga and Sasaki[39,40], Eilbracht[41], Xia and Ding[42], Leitner[43] and our group[44,45] reported $Ru_3(CO)_{12}$, $Rh(acac)(CO)_2$, and $[RhCl(CO)_2]_2$ as classic homogeneous catalysts for CO production from the RWGS reactions in the presence of additives, and subsequent transformations in specific carbonylations in autoclaves. High pressures of $CO_2$ and $H_2$ in varying ratios and high temperatures are essential. Skrydstrup and co-workers reported the utilization of silacarboxylic acids (synthesized from chlorosilanes and $CO_2$), or disilane compounds with $CO_2$ to produce CO in homogeneous system. The generated CO was then applied in Pd-catalyzed carbonylation reactions by using the two-chamber system. Electrochemically, only one example was reported by Skrydstrup, Daasbjerg, and co-workers employing electrocatalyst iron porphyrin for $CO_2$-to-CO conversion and incorporating produced CO into carbonylations also in two-chamber system. The major challenge in electrochemical transformation is to increase the CO generation rate[28,46,47]. Based on our long-standing interest in carbonylation reactions, herein we report a new heterogeneous copper catalyst enabling highly selective CO generation from $CO_2$ in a continuous mode. The advantages of our catalytic system are no need of adding additives, long-term stability, easy recycle, and enough CO generation rate. Moreover, this reaction can be directly combined/switched with different industrially relevant and synthetically useful carbonylation reactions under ambient conditions. The described overall cascade process allows an efficient use of $CO_2/H_2$ as safe and green CO surrogate with follow-up utilization (Fig. 1). It is also one kind of mini-plants that initially explores the application of "two-chamber system" (Fig. S1).

## Results and discussion
### Selective CO generation from $CO_2/H_2$ (step I)
Inspired by previous efforts to develop catalysts for CO generation from $CO_2$[17,48–53], selected novel Cu-based materials were prepared following precipitation hydrothermal (PHM) and conventional impregnation (CIM) methods (preparation details see SI 2.1). First, silica was selected as a support due to its inertness and role both as carrier and ligand[54,55]. Apart from that, copper nanoparticles on alumina and carbon were synthesized for comparison. All these materials were tested

for the hydrogenation of $CO_2$ to CO between 200 and 400 °C in a continuous flow reactor. As shown in Table 1, $SiO_2$ supported Cu nanoparticles exhibited superior $CO_2$ hydrogenation activity compared to $Al_2O_3$ and carbon black. Increasing the reaction temperature from 250 to 400 °C led to improved $CO_2$ conversion, while excellent CO selectivity was preserved (SI, Fig. S4a). No significant MeOH formation is observed throughout the studied temperature range, which is contrast to the standard methanol catalyst system ($Cu/Zn/Al_2O_3$) and related materials. However, a small amount of $CH_4$ (1–2%) was detected at the higher temperature. The conventionally prepared $Cu@SiO_2$ (Table 1, entry 5) showed negligible activity as compared to the catalyst prepared using PHM. The effect of reaction pressure in the continuous-flow system at 400 °C has no influence on the overall $CO_2$ conversion (SI, Fig. S4b).

### Catalyst characterizations
To understand the activity and the structural features of the different $Cu@SiO_2$ materials, detailed characterizations using ICP, BET, XRD, TPR, TEM, XPS, and CO-IR techniques were performed (SI, 2.2). The physiochemical properties including surface area, pore volume, and pore diameter were determined using BET analysis (SI, Table S1). All the samples had high specific surface area which decreased with increased metal loading (Table S1, entries 2–4) due to the blocking of the pores with metal particles. X-ray diffraction (XRD) pattern of $10Cu@SiO_2$-PHM revealed only the presence of $Cu_2O$ species, while the conventionally prepared $10Cu@SiO_2$-CIM showed the co-existence of different oxidized Cu species (CuO, $Cu_2O$). In addition, the hydrothermally prepared material exhibited smaller and well-dispersed Cu crystallites (SI, Fig. S2). For both materials also metal oxide reduction studies were carried out using hydrogen temperature programmed reduction ($H_2$-TPR) (SI, Fig. S3a–c). While the active catalyst sample displayed a single reduction peak for Cu species with uniform dispersion, the less active systems showed two peaks, which is in accordance with the phase composition determined by XRD. Transmission electron microscopy (TEM) images of the optimal $10Cu@SiO_2$-PHM material revealed the existence of copper nanoparticles in the range of 1 to 10 nm, partially agglomerated into bigger polycrystalline structures (Fig. 2a).

The Cu $2p_{3/2}$ spectrum (XPS analysis) of this sample is dominated by a broad peak between 933–934 eV and a weak satellite around 943 eV (Fig. 2b). The main peak was deconvoluted in two contributions centered at 932.5 and 933.8 eV. The binding energy (BE) values and the satellite, a typical feature for Cu(II), are ascribed to the presence of both Cu(I) and Cu(II) species[56]. Apparently, surface Cu(I) species are oxidized in air to some extent to Cu(II) species, which might be reduced again under reductive reaction conditions. Indeed, in case of a spent catalyst, the Cu $2p_{3/2}$ spectrum exhibited a main symmetric peak at 932.5 eV ($Cu^{0/I}$) with only very weak satellite. However, in the $10Cu@SiO_2$-CIM sample the component of the main peak is shifted to higher BE (934.6 eV), and it can be ascribed to a stronger interaction of Cu(II) with the support in agreement to the $H_2$-TPR analysis. In fact, the experimental amount of $H_2$ consumption of an inactive $10Cu@SiO_2$-CIM sample was 1290.8 μmol/g (ICP = 8.3 is 1307 μmol/g), indicating that $Cu^{2+}$ was almost completely reduced to $Cu^0$ at 390 °C. However, the most active $10Cu@SiO_2$-PHM showed 627.97 μmol/g (ICP = 9.1 is equal to 1433.1 μmol/g) $H_2$ consumption. This again proves the most active sample contains a mixture of $Cu^{1+}/Cu^{2+}$, while in the inactive sample we have only $Cu^{2+}$ according to the $H_2$ consumption. In addition, it should be noted that the active sample reduction profile starts at lower temperature in comparison with $10Cu@SiO_2$-CIM (SI, Fig. S3b, c).

Furthermore, in situ IR spectroscopic experiments were performed under reaction conditions ($CO_2/H_2$ at 400 °C) to monitor the progress of the reaction. As shown in Fig. 2c, a strong formation of CO is observed in the presence of $10Cu@SiO_2$-PHM, while a much lower

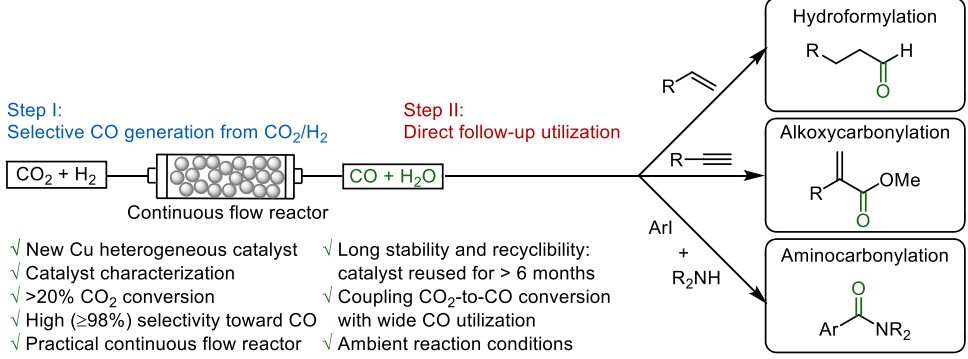

**Fig. 1 | Catalytic carbonylations using carbon dioxide.** Selective copper-catalyzed CO generation from $CO_2$ and direct utilization in carbonylation reactions.

**Table 1 | $CO_2$ hydrogenation to CO over different Cu-based catalysts in continuous-flow reactor[a]**

| Entry | Catalyst | $X_{CO2}$ [%] | $S_{CO}$ [mol. %] | $S_{MeOH}$ [mol. %] | $S_{CH4}$ [mol. %] |
|---|---|---|---|---|---|
| 1 | $5Cu@Al_2O_3$-PHM | ≤1 | 0 | 0 | Trace |
| 2 | 5Cu@C-PHM | 0 | 0 | 0 | Trace |
| 3 | $5Cu@SiO_2$-PHM | 11 | 96 | 3.6 | 0 |
| 4 | $10Cu@SiO_2$-PHM | 27 | 99 | Trace | Trace |
| 5 | $10Cu@SiO_2$-CIM | 5 | 98 | Trace | 2 |
| 6[a] | $10Cu@SiO_2$-PHM | 10.4 | 97 | 3 | Trace |

Reaction conditions: 300 mg of catalyst, 100 NmL/min of $H_2$/$CO_2$(3:1) gas mixture, gas hourly space velocity (GHSV) = 15,000 $h^{-1}$, temperature ($T$) = 400 °C, pressure ($P$) = 10 bar.
[a]Reaction temperature = 250 °C.

amount of CO is detected using $10Cu@SiO_2$-CIM, clearly indicating the higher activity of the former catalyst. Notably, for both catalyst materials excellent selectivity is observed as no side-products are identified. To proof further the nature of Cu species by in situ IR spectroscopy, additional CO adsorption studies at room temperature were performed. A strong band around at 2131 $cm^{-1}$ was detected for the $10Cu@SiO_2$-PHM sample, which can be assigned to $vCu^I$-CO and/or $vCu^0$-CO vibrations. In contrast, only a weak and broad band was detected in case of the $10Cu@SiO_2$-CIM sample. The different spectra again indicate a distinct higher concentration of $Cu^{0/I}$ species for the more active catalyst[57].

Finally, catalyst stability studies were carried out for selective $CO_2$ reduction to CO. As shown in Fig. 3, the optimal material showed no signs of deactivation after 100 h on-stream at 400 °C and 20 bar pressure ($CO_2$/$H_2$ = 1:3). The XPS analysis (Fig. 2b) of the spent catalyst indicate the presence of a dominant $Cu^+$ phase with no diffraction peaks attributing to $Cu^{2+}$ species, which is in accordance with the XRD analysis.

**Direct follow-up utilization (step II)**

After having a stable copper catalyst on silica for selective CO generation from $CO_2$ in hand, we then were curious about its direct utilization in carbonylations[16,58,59]. Notably, the following carbonylation reactions worked smoothly without tedious gas isolation/purification (no need to remove $CO_2$ and $H_2O$). Utilizing the described equipment (see SI), the recycling of the generated CO is not possible at present. However, by adding an additional reactor or by installing a gas compressor it should be also possible to recycle CO.

Initially, the metal-catalyzed homogeneous hydroformylation reaction was tested as it not only represents one of the most important industrial processes, but also constitutes an attractive synthetic method for producing aroma compounds, which are used as ingredients in numerous perfumes and flavors[60,61]. Utilizing 1-octene **1a** as the model substrate in this flow reaction mode we explored various catalytic systems and conditions. After the optimization (SI, Table S2), Rh(acac)(CO)$_2$/6-DPPon was defined as the best catalyst. To our delight, hydroformylation proceeded smoothly even under ambient

conditions and the linear product n-nonanal (pelargonaldehyde) **2a**, which has a fatty-rose-like odor and confers a typical rose nuance into floral perfumes, was obtained in 92% yield (Fig. 4a).

The shorter chain n-heptanal **2b** was synthesized in quantitative yield. The strong fruity odor makes it widely applied in the aroma industry, and due to its special scent in dilute solution or derived acetal, it is also an important ingredient in soap perfumes and domestic fragrances[60]. The longer chain olefin 1-dodecene **1c** is often converted only in moderate yield and l/b selectivity in conventional hydroformylation reaction[60]. Its linear product n-tridecanal **2c** has a slightly floral scent that is reminiscent of citrus and grapefruit peel. Gratifyingly, in this reaction system **2c** was obtained in 90% yield with excellent selectivity (>99:1). The hydroxy aldehyde **2d** is used as a main component in the production of a perfume with a smell like lily of the valley; however, industrially, it was synthesized in moderate yield from the hydroformylation of 3,7-dimethylocten-1-en-7-ol in benzene under high pressure (85 bar)[60]. In contrast, **2d** is easily attained in 98% yield at room temperature and ambient conditions here. Silyl aldehydes are valuable synthetic intermediates[62], which can be regarded as latent carbanions and therefore can be used as bifunctional reagents[63]. As an example, the silyl aldehyde **2e** is isolated in quantitative yield. Furthermore, this mild and convenient method is also compatible for the hydroformylation of aromatic olefins (allyl benzene **1f**) as well as bio-renewable derivatives (eugenol[64] **1g**). The corresponding aldehydes **2f** and **2g**, which are applied in, cosmetic pharmaceutical and food industries[65], were synthesized in good to excellent yields (79->99%). Additionally, styrene (**1h**) and its derivatives (**1i** and **1j**) were fully converted to the corresponding aldehydes (**2h**–**j**), which are isolated in excellent yields (94–99%) with the branched isomers as major products. However, using acrylates or acrylonitrile under the here reported mild conditions, no activity was observed. A list of the non-successful substrates can be found in the SI (Fig. S5).

Next, the methoxycarbonylation of alkynes was explored as another important method. Apart from many synthetic applications, acrylates resulting from propyne, arylacetylenes, and the parent compound are industrially important molecules as they serve as

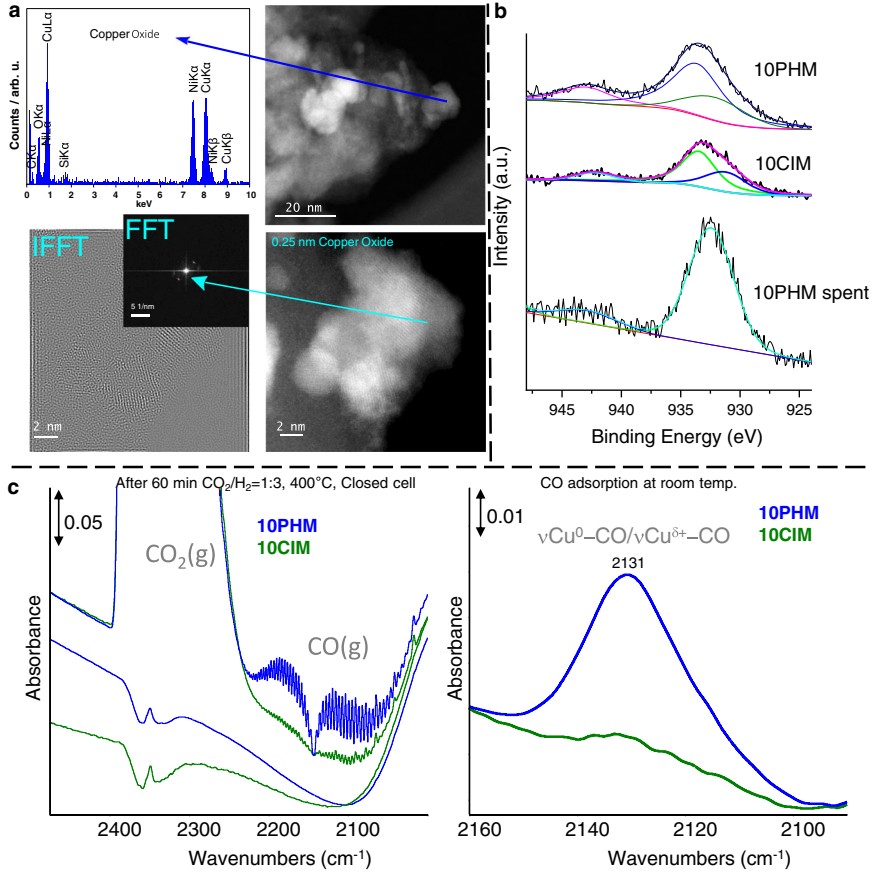

**Fig. 2 | Catalyst characterizations. a** STEM-HAADF images, EDXS and FFT/IFFT of fresh 10Cu/SiO₂-PHM sample. **b** XPS spectra of 10Cu@SiO₂-PHM, 10Cu@SiO₂-CIM, and 10Cu@SiO₂-PHM-spent samples. **c** In situ FTIR spectra of 10Cu@SiO₂-PHM and 10Cu@SiO₂-CIM catalysts obtained at 400 °C before and after 60 min exposure to CO₂/H₂ (left) and CO adsorbate spectra measured at room temperature after the in situ experiment (right).

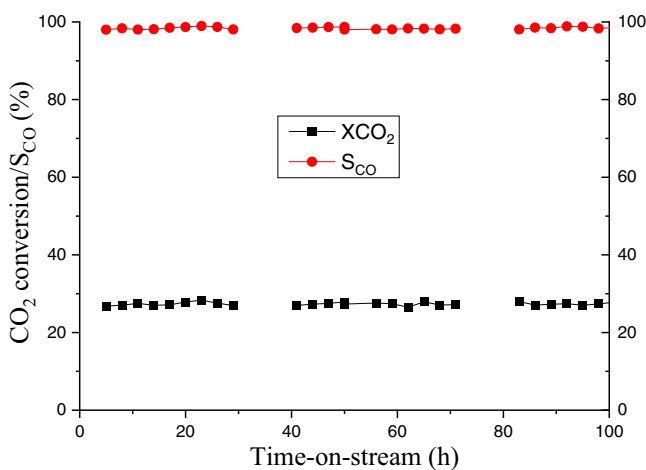

**Fig. 3 | Catalytic stability test for CO₂ conversion (XCO₂) and CO selectivity (SCO).** Reaction conditions: 300 mg of 10Cu@SiO₂-PHM, 100 NmL/min of H₂/CO₂ (3:1) gas mixture, gas hourly space velocity (GHSV) = 15,000 h⁻¹, temperature ($T$) = 400 °C, pressure ($P$) = 20 bar.

polymer precursors or fine chemicals[66]. For example, the global demand of methyl methacrylate (MMA) is over 4.8 million metric tons[67]. Thus, the methoxycarbonylation of propyne was developed by Shell in 1986[68] as an attractive one-step process to obtain MMA[68]. Applying specifically the combination of Pd(Oac)₂/PPh₂Py/MsOH as a state-of-the-art catalyst system, Drent et al. realized the synthesis of

MMA with an impressive TOF of 40,000 h⁻¹ at 45 °C under 60 bar of CO[69]. In this work, again the CO generated from CO₂ reduction was directly introduced in the mixture of alkynes and Pd catalyst at ambient conditions, which makes the overall system easy to use. After a brief optimization using phenylacetylene **3a** as the model substrate, the branched product **4a** was obtained in quantitative yield with >99% branched selectivity in the presence of the Drent catalyst system (Fig. 4b). To our delight, other substituted aryl derivatives **3b** as well as aliphatic substrates bearing chloro (**3c**) and cyano (**3d**) groups gave the corresponding acrylates **4b**, **4c**, and **4d** in good to excellent yields (85–>99%).

Finally, the palladium-catalyzed aminocarbonylation of aryl halides was explored using CO directly from carbon dioxide and hydrogen. Such carbonylations represent a valuable tool in organic synthesis, and the resulting benzamides are important structures in many pharmaceuticals, dyes, agrochemicals, and other industrial products[26,70]. Indeed, 4-iodoanisole **5a** and related arenes reacted smoothly with piperidine **6a** by using [Pd(dba)₂]/PPh₃ as the catalyst. After directly introducing the gas mixture products **7a**–**c** were isolated in 90–94% yield (Fig. 4c). Furthermore, linear secondary and primary amines were converted smoothly to the desired amides **7d** and **7e** were attained in good yields. Apart from the presented concept, this methodology is also noteworthy with respect to the selectivity because only monocarbonylated products could be observed, while related aminocarbonylations provided dicarbonylated products[71,72]. In addition, the high product yield at comparably low pressure is noteworthy.

Finally, it is worthwhile mentioning that the produced water from the RWGS reaction did not affect the investigated carbonylations here.

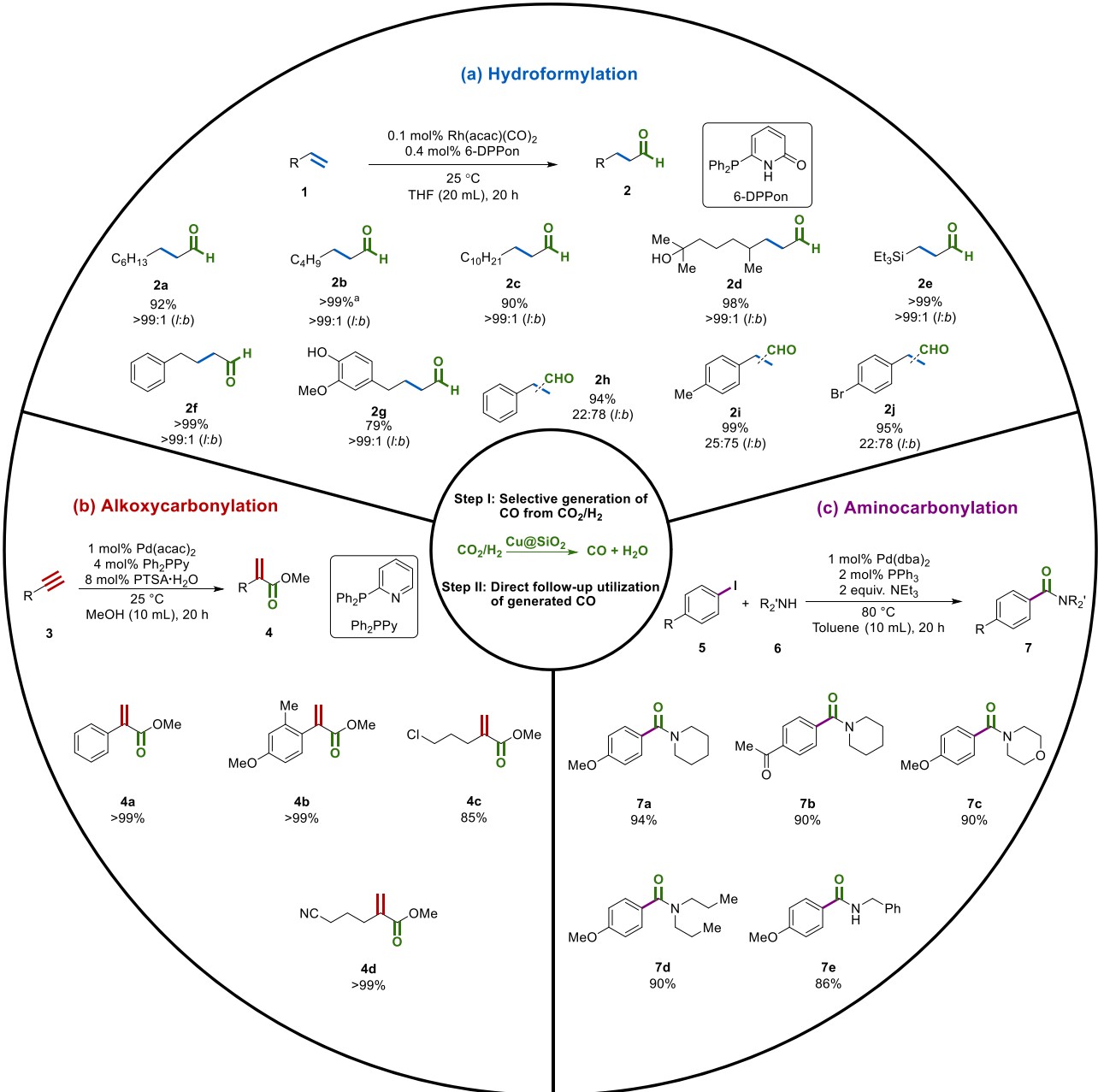

**Fig. 4 | Cu-catalyzed selective CO₂-to-CO conversion and the follow-up utilization in carbonylation reactions. a** Hydroformylation of alkenes. **b** Alkoxycarbonylation of alkynes. **c** Aminocarbonylation of aryl halides.

Moreover, despite the mild conditions, all the presented carbonylation reactions proceeded with similar rates compared to related carbonylations with homogeneous catalysts[61,69,73–76].

In conclusion, we present a practical concept for carbonylation reactions simply utilizing carbon dioxide and hydrogen as a safe source of carbon monoxide. Following our idea, CO is generated on demand and directly utilized in several carbonylation reactions. For the highly selective CO generating process, a novel hydrothermally prepared heterogeneous catalyst (Cu@SiO₂-PHM), which constitutes of uniformly dispersed Cu(I) species, proved to be optimal. The generated carbon monoxide is used in industrially relevant hydroformylations, alkoxycarbonylations, and synthetically interesting aryl halide carbonylations to produce a variety of functionalized carbonyl compounds in good yields and selectivities. Notably, all these transformations can be performed in a general manner under mild conditions (ambient pressure, low

temperature). Thus, this approach is of interest for continuous syntheses in drug discovery and organic synthesis to produce building blocks on reasonable scale, too. In general, the presented methodology offers a cost-effective and safe way to perform various carbonylation processes (hydroformylation, alkoxy- and aminocarbonylation) using CO₂ instead of CO.

## Methods

### Materials and characterization methods

Air- and moisture-sensitive synthesis were performed under argon atmosphere in heating gun vacuum dried glassware. Chemicals were purchased from Aldrich, TCI, Alfa, Fluka, Acros, or Strem. Unless the purity was <97%, all commercial reagents were used without further purification. Cu(NO₃)₂•3H₂O (≥99%) salt was obtained from Sigma-Aldrich. Aqueous NH₃ solution (28–30%) was purchased from Roth Chemicals. Silica 60 M was obtained from Macherey-Nagel Germany.

$CO_2$ (99.99%), $H_2$ (99.99%), Ar (99.99%) and $N_2$ (99.95%) were provided by Linde Europe. All solvents were degassed prior to use. The products were characterized by $^1H$ NMR and $^{13}C$ NMR spectroscopy. $^1H$ and $^{13}C$ NMR spectra were recorded on Bruker Avance 300 (300 MHz) or 400 (400 MHz) NMR spectrometers. Chemical shifts δ (ppm) are given relative to solvent: references for $CDCl_3$ were 7.27 ppm ($^1H$-NMR) and 77.00 ppm ($^{13}C$-NMR). $^{13}C$-NMR spectra were acquired on a broad band decoupled mode. Multiplets were assigned as s (singlet), d (doublet), t (triplet), dd (doublet of doublet), and m (multiplet). GC analysis for the gas phase was performed using Agilent HP-PLOT/Q fitted with TCD (thermal conductivity detector) and FID (flame ionization detector) detectors. Carbonylation-related GC analysis was performed on a Trace 1310 chromatograph with a 29 m HP5 column. The products were measured by MS and GC analysis or by isolation from the reaction mixture by solvent evaporation and further purified by column chromatography on silica gel. BET surface area and pore volume of the prepared catalysts were measured from nitrogen adsorption isotherms measured at −196 °C (Micromeritics ASAP 2010). Before the measurement, each sample was degassed at 200 °C for 4 h. The average pore diameters were calculated from the desorption branch of the isotherm using the BJH method. Inductively coupled plasma optical emission spectrometry (ICP-OES) analysis was performed using Varian/Agilent 715-ES analyzer. XRD powder patterns were recorded on a Stoe STADI P diffractometer, equipped with a linear Position Sensitive Detector (PSD) using Cu K radiation (λ = 1.5406 Å). Processing and assignment of the powder patterns was done using the software WinXpow (Stoe) and the Powder Diffraction File (PDF) database of the International Centre of Diffraction Data (ICDD). For the TPR experiments, the measurement was done using a Micromeritics Autochem II 2920 instrument. A 100 mg sample was loaded in U shaped quartz reactor and heated from RT to 400 °C with 20 K/min in 5% $O_2$/He (50 ml/min) for 30 min at 400 °C, then flushing and cooling down to RT under the flow of Ar. The TPR measurement was carried out from RT to 700 °C (holding time 30 min) in a 5% $H_2$/Ar flow (50 ml/min) with a heating rate of 10 K/min. Another TPR measurement was done as described previously only in the pretreatment step Ar was used rather than $O_2$. The hydrogen consumption peaks were recorded with temperature using thermal conductivity detector. Quantitative analysis of the TPR data was calculated based on the peak areas. The TEM measurements were performed at 200 kV with an aberration-corrected JEMARM200F (JEOL, Corrector: CEOS). The microscope is equipped with a JED-2300 (JEOL) energy-dispersive X-ray-spectrometer (EDXS) and an Enfinum ER (GATAN) with Dual EELS for chemical analysis. The solid samples were deposed without any pretreatment on a holey carbon-supported Ni-grid (mesh 300) and transferred to the microscope. XPS data were obtained with a VG ESCALAB220iXL (Thermo-Scientific) with monochromatic Al Kα (1486.6 eV) radiation. Binding energies were corrected to C−C contribution at 284.8 eV in C1s region. For quantitative analysis, the peaks were deconvoluted with Gaussian-Lorentzian curves, the peak area was divided by a sensitivity factor obtained from the element-specific Scofield factor and the transmission function of the spectrometer. For the characterization of the Cu species CO was used as probe molecule. In situ FTIR spectroscopic measurements in transmission mode were carried out on a Bruker Tensor 27 FTIR spectrometer equipped with a heatable and evacuable homemade reaction cell with $CaF_2$ windows connected to a gas-dosing and evacuation system. The sample powders were pressed into self-supporting wafers with a diameter of 20 mm and a weight of 50 mg. The samples were pretreated by heating in vacuum up to 400 °C and keeping at this temperature for 1 h. After dosing $CO_2$/$H_2$ = 1:3 for 5 min the reaction cell was closed, and the reaction was monitored for 60 min. After cooling to room temperature and evacuation the sample was exposed to 5% CO/He. The CO adsorbate spectrum was recorded after removing the gas phase by evacuation of the cell.

## Cu@supports-PHM catalysts preparation

The catalysts were prepared using precipitation-hydrothermal method (PHM). In a typical synthesis process, desired amount of metal precursor was dissolved in 20 mL of DI $H_2O$ to obtain different metal loadings and stirred for 15 min. 2.5 ml of 28–30% Aq. $NH_3$ solution was added dropwise under continuous stirring for 30 min. Then the support ($SiO_2$, $Al_2O_3$, or C) was added to the precipitate and the mixture was stirred vigorously for 4 h at room temperature. The contents were then transferred to a 100 mL autoclave and heated at 120 °C for 8 h without stirring. The autoclave was then allowed to cool naturally to room temperature and the slurry was centrifuged and washed several times with ethanol and dried at 80 °C for 10 h. The dried material was then pyrolysed at certain temperature for 2 h under Ar atmosphere.

## Step I, selective CO generation from $CO_2$/$H_2$

The reaction for reducing $CO_2$ with $H_2$ into CO was carried out in a fixed bed flow-reactor (8.9 mm ID). The catalyst bed at the center of the reactor was supported by glass wool and quartz sand (dilution 1:5). Temperature was measured by a K-type thermocouple inserted into the catalyst bed. A gas mixture of $H_2$ and $CO_2$ ($H_2$:$CO_2$ = 3:1) at a total flow rate of 100 NmL/min was fed into the reactor (300 mg catalyst, 0.25–0.42 mm, GHSV = 15,000 $h^{-1}$). The volumetric flow rate of the feed gases was controlled by pre-calibrated mass flow controllers (Brooks Instrument). The reaction temperature was increased from 200 to 400 °C. The $CO_2$ conversion and product selectivity are defined as follows:

$$\text{Conversion} = 100 \times m\text{CO}_{2(in)} - m\text{CO}_{2(out)}/m\text{CO}_{2(in)} \qquad (1)$$

$$\text{Selectivity} = 100 \times m_{\text{product}_{(out) \times \text{carbon number}}}/m\text{CO}_{2(in)} - m\text{CO}_{2(out)} \qquad (2)$$

where $m\text{CO}_{2(in)}$ and $m\text{CO}_{2(out)}$ are the moles of $CO_2$ in and out of the reactor. The selectivity is defined as the percentage of moles of $CO_2$ consumed to form desired product (CO, $CH_4$, or MeOH), in respect to the amount of $CO_2$ consumed during the reaction.

## Step II, direct follow-up utilization

Under argon atmosphere, a 25 mL three-necked round bottom flask was charged with certain catalyst. Ligand and additives, as well as a stirring bar. The flask was assembled with a −20 °C condenser and a pipeline which connect to the fixed bed flow-reactor. The water generated during the $CO_2$ reduction (step I) was not removed and the resulting gas mixture ($CO_2$, $H_2$, CO) was constantly bubbled through the reaction solution at flow rate of 100 ml/min. Substrate and solvent were injected into the flask by syringe. The flask was then sealed with cap, the condenser was connected to the ventilation system and the flow-reactor was opened for bubbling in the solution. The GC analysis was performed while taking a gas sample directly from the reaction flask to get the gas composition (%). The reaction was performed for 20 h at 25 °C. After the reaction finished, the l/b selectivity was determined by GC analysis using isooctane as the internal standard. The yields were isolated yields after purification with the column chromatography.

## Data availability

All data generated or analyzed during this study are included in the published article or its Supplementary Information data files.

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

## Acknowledgements

The authors acknowledge financial support from the State of Mecklenburg-Vorpommern, European Union, CADIAC, the iBOF (Next-BIOREF) and the EoS FWO/FNRS BioFact project (No. 30902231), for funding. B.U.W.M. thanks the Francqui Foundation for an appointment as Collen-Francqui professor. We thank Mr. Reinhard Eckelt for BET measurements, Mrs. Marga-Martina Pohl, Nils Rockstroh, Giovanni Agostini for TEM-XPS analysis, and Dr. Henrik Lund for the XRD results.

## Author contributions

R.S., Y.H., and R.R. contributed equally to this work. R.S., R.R., Y.H., R.J., H.J., and M.B. conceived and designed the experiments. R.R., R.S., and R.J. built the follow chemistry system and reaction equipment. R.R., R.S., H.N., and R.J. prepared the catalysts. R.S., R.R., Y.H., and G.M. performed the experiments and analyzed the data. H.A., and U.B. performed TPR and in situ IR analyses. R.S., R.R., Y.H., M.S., B.M., and M.B. co-wrote the paper.

## Funding

## Competing interests

The authors declare no competing interests.
