## [Peer Review File · Nature Communications]

Title: A practical Concept for Catalytic Carbonylations using Carbon DioxideREVIEWER COMMENTS

Reviewer #1 (Remarks to the Author): 
comments attached.

Reviewer #2 (Remarks to the Author):

The manuscript provides an interesting approach to make carbonylation products from in-situ generated CO from CO₂ hydrogenation. The work is novel in terms of the method used for carbonylation reactions and also all the three different reactions worked well with this method. The catalyst designed for CO₂ hydrogenation is also prepared by a new method which enhanced performance and sufficient characterizations are reported. The following comments need to be addressed properly.

1) What happens to the water generated during the RWGS reaction? Was it removed before passing the out gas from the first reactor to the round bottom flask for the cascade reaction? It needs to be mentioned clearly.

2) Instead of weight hourly space velocity, better to represent it in Gas hourly space velocity (in h⁻¹ unit) which is more appropriate for gas phase reactions.

3) Conclusion: "...($\text{Cu@SiO}_2\text{-PHM}$), which constitutes of uniformly dispersed Cu(I) species, proved to be optimal....." If Cu(I) species is present in the catalyst, it should have shown in H₂-TPR but there was only one CuO peak in $10\text{Cu@SiO}_2\text{-PHM}$. What's your explanation?

4) As the out gas from the first reactor was directly passed for the subsequent reaction, Out gas composition from the first reactor becomes very important. As the CO₂ conversion is only 27%, there is a large amount of CO₂ passing to the subsequent reaction, which is overlooked in all your schemes and explanations. Secondly, how much hydrogen remained after reacting with CO₂ which is then required for hydroformylation etc reactions? Hence, the feed composition (CO+H₂+CO₂ with appropriate %) for the second reaction needs to be provided for all the reactions studied here.

5) "The flow-reactor was opened for bubbling in the solution." This was followed for all the hydroformylation etc reactions. What was the bubbling rate? This needs to be mentioned. As the reaction was carried out for 20 h with bubbling of CO+CO₂+H₂ mixture to convert mere 10 mmol of a substrate, the reaction rate appears to be very slow. Hence, the reaction results of previously reported catalysts need to be compared along with reaction conditions to get an idea how good this cascade method is.

6). Line 81. "Indeed, only a few chemical and electrochemical CO₂-to-CO conversion coupled with CO utilization for carbonylations were reported, until now." This needs to be elaborated with specific reported examples and explain properly how different is your method reported in this manuscript because the major novelty of your work lies here.

7) "In general, the presented methodology offers a cost-effective and safe way to perform all kinds of carbonylation processes using CO₂ instead of CO" This last sentence of your conclusion is a tall claim. Inherent dilution of CO and H₂ with CO₂ is unavoidable and I am not sure this in-situ generated CO

along with CO₂ will perform well for all kinds of carbonylation processes. So, better to stick to the reactions studied in this work only and not try to generalize.

Response to referees

Reviewer 1:

Remarks to the Author:

In this manuscript, the authors demonstrated a practical concept for catalytic carbonylations using carbon dioxide. They developed a novel heterogeneous copper catalyst 10Cu@SiO₂-PHM, which showed high selectivity and decent conversion in generating CO from reducing CO₂ with H₂. The generated CO from a continuous flow reactor can be directly utilized in several carbonylation reactions to construct a variety of aldehydes, acrylates, and amides in high yields. These transformations can be performed in a general manner under mild conditions, which provide possibility for further industrial application. In summary, the developed methodology offers a cost-effective and safe way to perform several kinds of carbonylation processes using CO₂ instead of toxic CO. Therefore, I think this work should be suitable to be published on Nature Communications after the following revisions:

1. In the general procedures for carbonylation reactions, the generated CO from a continuous flow reactor is bubbling in the reaction solution during the reaction time. From a practical point of view, how much CO is consumed for each reaction? Can the excessive CO be recycled or not? Is there any information for the actual pressure in the carbonylation reaction during the reaction time?

Our response:

We thank the referee for the positive assessment of our work and the general questions. Utilizing the described equipment, the recycling of the excessive CO is not possible. However, by adding an additional reactor or by installing a gas compressor CO should be recyclable. At present, in our system the step II (carbonylation reaction) is performed at atmospheric pressure and the reactor is directly connected to the ventilation system for the out gas via gas tube. To make this clearer, a sentence has been added into the manuscript on page 6.

2. In Scheme 2, the substrate scopes of alkenes and alkynes may need to be further extended. In the manuscript, the authors considered allyl arene (1f and 1g) as aromatic olefins, which might be not accurate. The authors should investigate the carbonylation of real aromatic olefins, such as styrenes, and electron-deficient alkenes, such as acrylates and acrylonitrile. In addition, the reaction of disubstituted alkynes should also be investigated.

Our response:

As suggested by the referee we performed the reaction with additional substrates. Utilizing styrene and its derivatives, good reactivities were observed, no matter with electro-withdrawing (Br-) or electro-donating group (Me-). However, using acrylates, acrylonitrile or disubstituted alkynes under the here reported conditions, no activity was observed. A discussion on these reactions can be found at page 8 and the newly prepared products are shown in Scheme 2 (**2h-j**).

3. In the introduction, more reviews about transformation of CO₂ (such as Green Chem. 2017, 19, 3707-3728; Acc. Chem. Res. 2021, 54, 2518) should be cited. And some related reviews about carbonylation using CO₂ (such as Chin. J. Chem. 2018, 36, 353-362; Chem. Commun. 2020, 56, 8355-8367) should also be cited.

Our response:

Thank you for your kind suggestion. All the references were added to the reference section.

4. There are some errors in the Supporting Information, such as redundant symbols in the ¹H NMR of 2c and 2f. And the HRMS(EI) data of 7d and 7e should be HRMS(ESI) data. Additionally, please double check the ¹³C NMR of 7b, 7c and 7d carefully.

Our response:

We thank the referee for the careful reading and apologies for the mistakes. All mentioned errors were corrected. In addition, we double checked the SI again.

Reviewer 2:

Remarks to the Author:

The manuscript provides an interesting approach to make carbonylation products from in-situ generated CO from CO₂ hydrogenation. The work is novel in terms of the method used for carbonylation reactions and also all the three different reactions worked well with this method. The catalyst designed for CO₂ hydrogenation is also prepared by a new method which enhanced performance and sufficient characterizations are reported. The following comments needs to be addressed properly.

1. What happens to the water generated during the RWGS reaction? Was it removed before passing the out gas from the first reactor to the round bottom flask for the cascade reaction? It needs to be mentioned clearly.

Our response:

We thank the referee for the positive assessment of our work and this question. The water generated during the RWGS reaction is not removed prior to the follow-up reaction setup. To make this clearer to the reader it is mentioned in the ESI (Figure SI and SI 3.1.1) and Schemes 1-2 have been revised accordingly. A sentence is also added in the manuscript on page 6 and page 9.

2. Instead of weight hourly space velocity, better to represent it in Gas hourly space velocity (in h⁻¹ unit) which is more appropriate for gas phase reactions.

Our response:

As suggested by the referee the gas hourly space velocity (in h⁻¹ unit) is now mentioned in the manuscript. A GHSV of 15.000 h⁻¹ has been achieved here (see Table 1, Figure 2 and SI 2.3).

3. Conclusion: “....(Cu@SiO₂-PHM), which constitutes of uniformly dispersed Cu(I) species, proved to be optimal.....” If Cu(I) species is present in the catalyst, it should have shown in H₂-TPR but there was only one CuO peak in 10Cu@SiO₂-PHM. What’s your explanation?

Our response:

We thank the referee for raising this point. Thus, we further investigated the catalyst samples by H₂-TPR analysis. The new measurements were done without any O₂ pretreatment instead the pretreatment was done in the presence of Ar at temperatures up to 200 °C. Then, the temperature was kept constant for 30 min at 200 °C. After cooling down the sample to room temperature, TPR was performed by heating the sample under the flow of 5% H₂/Ar, 10 K/min

up to 700 °C and kept for 30 min at 700 °C. According to the new TPR analysis, the experimental amount of H₂ consumption of an inactive 10Cu@SiO₂-CIM sample was 1290.8 μmol/g (ICP = 8.3 is 1307 μmol/g theoretical value), indicating that it contains mostly Cu²⁺ which was almost completely reduced to Cu⁰ at high temperature (390°C). However, our active 10Cu@SiO₂-PHM showed 627.97 μmol/g (ICP = 9.1 is equal to 1433.1 μmol/g theoretical value) H₂ consumption. This means the most active sample contains a mixture of Cu¹⁺/Cu²⁺, while in the inactive sample we have mostly Cu²⁺ with only 1% Cu¹⁺ according to the H₂ consumption. Additionally, the active sample reduction profile showed a lower reduction temperature at (104 °C) in comparison with 10Cu@SiO₂-CIM at (180 °C). Moreover, other supporting analysis (XRD/XPS, IR) also indicate the presence of Cu₂O species on the surface of the catalyst. This information is added into the manuscript on page 4 and SI 2.2.3.

4. As the out gas from the first reactor was directly passed for the subsequent reaction, Out gas composition from the first reactor becomes very important. As the CO₂ conversion is only 27%, there is a large amount of CO₂ passing to the subsequent reaction, which is overlooked in all your schemes and explanations. Secondly, how much hydrogen remained after reacting with CO₂ which is then required for hydroformylation etc reactions? Hence, the feed composition (CO+H₂+CO₂ with appropriate %) for the second reaction needs to be provided for all the reactions studied here.

Our response:

Thank you for raising this question. Indeed, there is a significant amount of CO₂ and H₂ passing to the subsequent reaction. This information is added on page 6. In the first reaction step, we were mainly interested to achieve a high selectivity for CO as CO₂ and H₂ can be in principle recycled after the second step. Notably, the following carbonylation reactions worked smoothly without tedious gas isolation/purification (no need to remove CO₂). To illustrate this better Scheme 2 has been modified accordingly.

As shown in Figure S1, CO was continuously generated from CO₂/H₂ passing through the copper catalyst and the feed composition was determined by GC analysis. The GC analysis was performed while taking a gas sample directly from the reaction flask (step II). The gas composition of the gas mixture is H₂ : CO : CO₂ = 70 : 9.0 : 21.

GC spectra:

5. “The flow-reactor was opened for bubbling in the solution.” This was followed for all the hydroformylation etc reactions. What was the bubbling rate? This needs to be mentioned. As the reaction was carried out for 20 h with bubbling of CO+CO₂+H₂ mixture to convert mere 10 mmol of a substrate, the reaction rate appears to be very slow. Hence, the reaction results of previously reported catalysts needs to be compared along with reaction conditions to get an idea how good this cascade method is.

Our response:

The reaction gas mixture from step I (CO₂, H₂, CO) was bubbled through the follow-up reaction flask at a flow-rate of 100 ml min⁻¹. Typically, reactions were carried out for 20 h at room

temperature (hydroformylations and alkoxycarbonylations). In case of aminocarbonylations 80 °C was used as reaction temperature. In most cases, the desired products were obtained in good yields and high selectivity under such mild reaction conditions. Obviously, the reaction rate for the second step can be simply increased by working at higher temperature. To better compare the results with previous works, additional references for these transformations under mild conditions have been added on page 3 and page 9.

6). Line 81. “Indeed, only a few chemical and electrochemical CO₂-to-CO conversion coupled with CO utilization for carbonylations were reported, until now.” This needs to be elaborated with specific reported examples and explain properly how different is your method reported in this manuscript because the major novelty of your work lies here.

Our response:

Thanks a lot for the good suggestions. Specific reported examples were elaborated, and the differences were briefly stated in the manuscript on page 3.

7) “In general, the presented methodology offers a cost-effective and safe way to perform all kinds of carbonylation processes using CO₂ instead of CO” This last sentence of your conclusion is a tall claim. Inherent dilution of CO and H₂ with CO₂ is unavoidable and I am not sure this in-situ generated CO along with CO₂ will perform well for all kinds of carbonylation processes. So, better to stick to the reactions studied in this work only and not try to generalize.

Our response:

We thank the referee for this comment. We have changed the text in the manuscript accordingly.

REVIEWERS' COMMENTS

Reviewer #1 (Remarks to the Author):

The authors here demonstrated a practical concept for catalytic carbonylations using carbon dioxide. According to the reviewer's suggestions, the authors carried out more experiments to explain and answer corresponding questions. The manuscript and Supporting Information have been well revised based on the reviewer's comments, so I recommend it is now suitable for publication in Nature Communications without any changes.

Reviewer #2 (Remarks to the Author):

All the comments and suggestions from the reviewers are properly addressed and the manuscript is thoroughly revised with additional experiments and literature addition. I am happy with the revision.